# Reconstructing Mayotte 2018–19 Rift Valley Fever outbreak in humans by combining serological and surveillance data

Jonathan Bastard [1]✉, Guillaume André Durand[2,3], Fanny Parenton[1], Youssouf Hassani[1], Laure Dommergues[4], Juliette Paireau[1,5], Nathanaël Hozé [5], Marc Ruello[1], Gilda Grard[2,3], Raphaëlle Métras [6] & Harold Noël [1]

## Abstract

**Background** Rift Valley Fever (RVF) is a zoonosis that affects large parts of Africa and the Arabian Peninsula. RVF virus (RVFV) is transmitted to humans through contacts with infected animals, animal products, mosquito bites or aerosols. Its pathogenesis in humans ranges from asymptomatic forms to potentially deadly haemorrhagic fevers, and the true burden of human infections during outbreaks is generally unknown.

**Methods** We build a model fitted to both passive surveillance data and serological data collected throughout a RVF epidemic that occurred in Mayotte Island in 2018–2019.

**Results** We estimate that RVFV infected 10,797 (95% CrI 4,728–16,127) people aged ≥15 years old in Mayotte during the entire outbreak, among which only 1.2% (0.67%–2.2%) were reported to the syndromic surveillance system. RVFV IgG seroprevalence in people ≥15 years old was estimated to increase from 5.5% (3.6%–7.7%) before the outbreak to 12.9% (10.4%–16.3%) thereafter.

**Conclusions** Our results suggest that a large part of RVFV infected people present subclinical forms of the disease and/or do not reach medical care that could lead to their detection by the surveillance system. This may threaten the implementation of exhaustive RVF surveillance and adequate control programs in affected countries.

## Plain language summary

Rift Valley Fever (RVF) is a disease caused by a virus transmitted from livestock animals to humans by mosquito bites, aerosols or direct contact with infected animals or animal products. In some parts of Africa and the Arabian Peninsula, the virus can lead to large outbreaks in both humans and animals. Despite some infected people developing severe forms of the disease, some experience no or mild symptoms. Therefore, infection is often not detected by surveillance systems based on the reporting of symptoms by patients. Here, we use data collected during a RVF outbreak that occurred in 2018–2019 in Mayotte Island, in the Indian Ocean, to model the course of the outbreak in humans. We estimate that, throughout the epidemic, only 1.2% of infected people were detected by the surveillance system. Our results highlight that most human cases may go unreported during RVF outbreaks, making it difficult to monitor the burden of infections.

[1] Santé publique France, French national public health agency, F-94415 Saint-Maurice, France. [2] French Armed Forces Biomedical Research Institute, National Reference Laboratory for Arboviruses, Marseille, France. [3] Unité des Virus Émergents (UVE: Aix-Marseille Univ-IRD 190-Inserm 1207), Marseille, France. [4] Groupement de Défense Sanitaire 976, Coconi, Mayotte. [5] Mathematical Modelling of Infectious Diseases Unit, Institut Pasteur, Université Paris Cité, UMR2000, CNRS, Paris, France. [6] Sorbonne Université, INSERM, Institut Pierre Louis d'Épidémiologie et de Santé Publique (IPLESP, UMRS 1136), Paris, France. ✉email: jb4753@columbia.edu

Rift Valley Fever (RVF) is a viral mosquito-borne disease affecting both food-producing animals and humans, reported in most parts of Africa and the Arabian Peninsula. In some regions, the enzootic reservoir of Rift Valley Fever virus (RVFV) may consist of domestic or wild animals[1,2]. Following particular environmental conditions (such as heavy rains) and/or introduction to new geographical areas, the virus can then cause large epizootics in food-producing animals, especially ruminants, and may result to numerous spill-over human cases (of all ages) infected by animals via mosquito bites, contacts with infected animals or animal products, or aerosols[2–5]. Estimating the burden of RVF epidemics in animals and humans is important to improve disease surveillance and control.

In animals, RVF can have serious health and economic impacts, causing high mortality and morbidity (including abortions) in livestock animals, and trade bans on live animals and animal products in affected countries[6]. In humans, RVF symptoms most often range from asymptomatic to dengue-like forms (febrile illness, myalgia, arthralgia) following an incubation time of 2–6 days[2,7,8]. But they can in some occasions evolve into more severe forms, such as encephalitis, hepatitis or a haemorrhagic syndrome sometimes leading to death[2,9]. Yet, the full impact of RVF epidemics on human health has rarely been assessed. Indeed, human RVFV infections are probably under-reported because (i) a proportion of human cases are subclinical, (ii) RVF symptoms are not specific and can be unrecognized, and (iii) RVF often occurs in countries with a poor access to healthcare and/or a poor surveillance system[3]. In this context, mathematical and statistical models can be of interest to investigate the true burden of infection by combining incomplete surveillance data with other sources of data[10,11]. In particular, serological data have the advantage to provide biological markers of both symptomatic and asymptomatic previous infections. In previous studies led in various areas of Africa, of the Indian Ocean and of Western and Southern Asia, serological data have been used to determine the proportion of a population exposed to RVFV in the past[3,12], to investigate the factors associated with such exposure[13–15], and to model epidemic dynamics[1,16,17].

Mayotte Island is an overseas region of France located in the Indian Ocean and populated by ~260,000 inhabitants[18]. In 2011, following a RVF outbreak in livestock in 2008–2010[19], a serological survey estimated the RVFV IgG seroprevalence in the human population of Mayotte (over 5 years old) to be 3.5%[15]. From late 2018 to mid-2019, the island experienced a RVF outbreak in animals and humans[16,20]. At the same period, between December 2018 and May 2019, the French public health agency conducted a seroprevalence study in the human population of Mayotte as part of a larger health survey (Unono Wa Maore survey)[21], thus providing a unique opportunity to estimate RVFV pre- and post-epidemic seroprevalence, and to quantify the completeness of RVF surveillance in humans.

Here, we developed a model combining surveillance data and serological data. We estimated that (i) RVFV seroprevalence in humans increased from 5.5% before the 2018–2019 epidemic to 12.9% thereafter, and that (ii) 1.2% of RVFV human infections were reported to the surveillance system during the outbreak.

## Materials and methods
We used two sources of RVF data collected during the 2018–19 epidemic: incident human cases as part of the passive surveillance system and serological data.

**RVF surveillance system in humans**. In Mayotte Island, patients with dengue-like symptoms generally first take a rapid malaria diagnostic test. If negative, they get a multiplex real-time reverse transcription PCR (RT-PCR) test for dengue, chikungunya and RVF viruses, as well as for *Leptospira*. This system has been in place since 2008. Information collected from reported cases include the date of birth, commune of residence, date of symptoms onset and date of RT-PCR confirmation. No RVF human case has been confirmed by RT-PCR on the island between 2009 and 2018.

**Serological data collection**. Throughout the outbreak timeline, RVF serological data were obtained from serum collected as part of the Unono Wa Maore study, a health survey led in a representative sample of the general population of Mayotte, described in[21]. Briefly, dwellings were randomly drawn from the register of localized buildings, a database containing housing addresses[22]. Within each participating household, up to three persons aged over 15 years old were then randomly selected. The characteristics of the surveyed population, notably in terms of age and sex, were comparable to the 2017 Mayotte general census[21].

For the RVF survey, 2853 blood samples collected between week 2018–49 (December 2018) and week 2019–21 (May 2019) in people aged between 15 and 69 years old were tested for RVFV IgG antibodies (Fig. 1 and Supplementary Data 1). These antibodies have been reported to be increasingly detectable from 8–10 days after RVFV infection in some previous publications[2,7,23], or from 6–17 days after symptoms onset in others[24,25]. Samples were tested by the National Arbovirus Reference Center. A homemade indirect ELISA was performed, using whole inactivated virus (Tchad 2001) and a goat anti-human IgG conjugated with peroxidase (Jackson Immuno Research, UK)[26]. Ratios of optical density (OD) between wells coated with inactivated virus and wells coated with negative antigen were calculated. As previously described[26–28], samples whose OD ratio was >3 were reported as IgG positive. A sensitivity analysis on the value of this cut-off was then performed (see below).

Due to the starting of the Ramadan, the acceptance of blood sampling as part of the Unono Wa Maore survey decreased from week 2019–18 (early May 2019) onwards, compromising the representativeness of the sampled population. As a result, we chose to analyze in this paper the serological data collected prior to week 2019–18, totalling 2566 samples.

**Ethics statement**. Unono Wa Maore research protocol was validated by the Committee for the Protection of Persons (CPP, no. 2017-A02782-51), the French ethical committee for biomedical research, and complied with MR001 reference methodology (agreement from the National Commission for Informatics and Freedoms of 25 September 2018, no. 918233). Information on the survey objectives and consent forms were read with the participants. A written informed consent was obtained from participants, or from a legal representative when participants were ≤17 years old. All methods were carried out in accordance with relevant guidelines and the Declaration of Helsinki. All samples and data were anonymized at the time of collection. Therefore, sample testing and data analysis were conducted anonymously.

**Study area**. We stratified the analysis by considering two sub-populations: people living in Central and Outer communes of Mayotte, as represented in Fig. 1 and as defined in Supplementary Note 1. This classification results from a previous publication[29] that analysed the characteristics of the livestock movement network across the island. Central communes exhibited a more intense movement pattern than Outer communes, which affected the spread of RVFV in the livestock populations of Mayotte and the spill-over to humans[29,30].

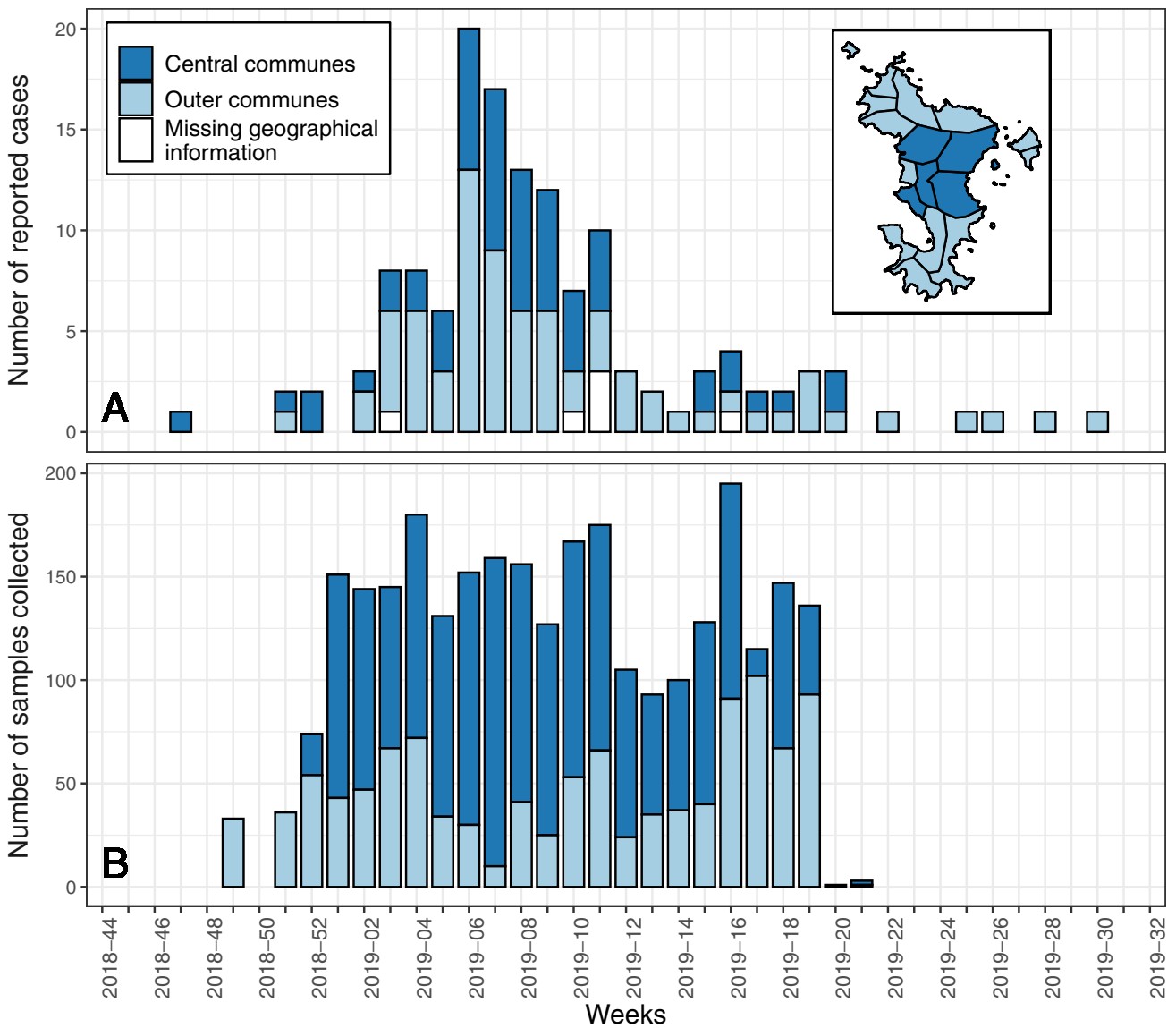

**Fig. 1 RVF data collected between week 2018–47 (November 2018) and week 2019–30 (July 2019) in people aged over 15 years old. A** Number of reported human cases by week from the surveillance system, and (**B**) number of blood samples collected by week as part of Unono Wa Maore survey. Light blue represents data collected from people living in the Outer communes of Mayotte Island, and dark blue in the Central communes, as depicted on the map at the top-right corner (see Supplementary Note 1 for details). For reported cases (**A**), we represent the week of symptoms onset or, if missing, the week of RT-PCR confirmation. For the seroprevalence study (**B**), we represent the week of sampling. Geographical information was missing for six reported cases aged over 15 years old.

### Statistics and reproducibility

*Model.* We developed a model to estimate RVFV attack rate and IgG seroprevalence in humans aged over 15 years old, during the course of 2018–2019 Mayotte outbreak, in two subpopulations i, determined by their place of residence (Central or Outer Communes)[29].

First, we modelled the epidemic curve using a lognormal function $F_i(t)$, defined as the weekly number of incident human infections (both reported and unreported to the surveillance system) in subpopulation i on week $t$:

$$F_i(t) = \frac{p_{1,i}}{t.\sqrt{2.\pi.p_{3,i}}} e^{\frac{-(log(t)-p_{2,i})^2}{2p_{3,i}}} \qquad (1)$$

where $t$ represents weekly time steps from week 2018–41 (October 2018) to week 2019–40 (October 2019). $p_{1,i}$ corresponds to the total number of people infected during the outbreak in subpopulation i. $p_{2,i}$ and $p_{3,i}$ determine the shape of the epidemic

curve including its duration, its starting date (i.e. the week $t$ for which $F_i(t) \geq 1$) and the date of its peak (i.e. the mode of the distribution). The three parameters were estimated from the data (Table 1).

Second, $I_i(t)$ was the number of infections on week $t$ in subpopulation i that were detected by surveillance (reported cases). $I_i(t)$ was assumed to follow a binomial distribution:

$$I_i(t) \sim Bin(\tau, F_i(t)) \qquad (2)$$

where $\tau$ was estimated from the data and represents the reporting fraction, i.e. the proportion of overall human infections that were reported to the surveillance system (Table 1). $\tau$ was assumed to be constant over the course of the epidemic and similar in all subpopulations. Because both RVFV incubation time[2,7,8] and viremia[2,31–34] are less than a week, we considered that the week of infection was the week of symptoms onset reported in the

**Table 1 Description of model parameters: notation, description, unit and value (estimated from data or extracted from the literature).**

| Parameter | Description | Unit | Value |
|---|---|---|---|
| $p_{1,i}$ | Number of people over 15 years old in subpopulation i infected during the outbreak (parameter of $F_i$ lognormal distribution) | – | Estimated |
| $p_{2,i}$ | Parameter of $F_i$ lognormal distribution (determines the shape of the epidemic curve) | – | Estimated |
| $p_{3,i}$ | Parameter of $F_i$ lognormal distribution (determines the shape of the epidemic curve) | – | Estimated |
| $\tau$ | Reporting fraction | – | Estimated |
| $S_{0,i}$ | IgG seroprevalence in subpopulation i before the outbreak | – | Estimated |
| $D$ | Time between infection and IgG detectability | Weeks | Estimated |
| $N_{Mayotte}$ | Number of people over 15 years old living in Mayotte | – | 144,262[18] |
| $N_{Central}$ | Number of people over 15 years old living in Central communes of Mayotte | – | 68,189[18] |
| $N_{Outer}$ | Number of people over 15 years old living in Outer communes of Mayotte | – | 76,073[18] |

More details on parameters' prior distributions are in Fig. 2 and Supplementary Table S1.

surveillance data or, when missing, the week of RT-PCR confirmation.

Third, we modelled $S_i(t)$, the RVFV IgG seroprevalence over time:

$$S_i(t) = S_{0,i} + \frac{\sum_{w=1}^{t-D} F_i(w)}{N_i} \qquad (3)$$

where $S_{0,i}$ was IgG seroprevalence in subpopulation i prior to the outbreak, D was the delay between the infection of an individual and the detectability of IgG antibodies in their blood, and $N_i$ the subpopulation size. Here, $S_{0,i}$ and D were estimated from the data as well (Table 1).

Finally, the weekly number of IgG positive samples in subpopulation i, $_iP(t)$, was modelled as:

$$P_i(t) \sim Bin(S_i(t), T_i(t)) \qquad (4)$$

with $T_i(t)$ being the number of individuals sampled on week $t$ in subpopulation i.

*Model fitting.* The model was fitted to the case count data and serological data using a Markov Chain Monte Carlo (MCMC) algorithm, implemented with R version 4.0.3 and *rjags* package[35]. The log-likelihoods of the "number of reported cases" and "number of IgG positive samples" components of the model for all weeks and for both geographical areas were summed together. Three chains were run for 200,000 iterations each, and every 200th value was sampled. For each chain, a burn-in of 150 samples was removed, as it was enough to allow the convergence of Markov chains. The effective sample size was at least 2340 for all parameters. Autocorrelation in the Markov chains was checked. The Gelman-Rubin statistic was below 1.2 for all parameters.

Estimated parameters are summarized in Table 1. We used non informative priors for most parameters (Fig. 2 and Supplementary Table S1). The prior of $p_{2,i}$ was set in order to search the mode of $F_i$ lognormal distribution (i.e. the true epidemic peak) between week 2018–50 (i.e. $t = 10$, December 2018) and week 2019–18 (i.e. $t = 30$, May 2019). This is why its prior distribution was uniform between $p_{3,i} + \log(10)$ and $p_{3,i} + \log(30)$. Moreover, $p_{1,i}$, the total number of people in subpopulation i infected over the course of the outbreak could not exceed $N_i$, the size of this subpopulation (Supplementary Table S1).

We then simulated the fitted model by computing 5000 repetitions, each of them using a different set of parameters randomly selected from the posterior chains.

*Sensitivity analyses.* We performed additional independent analyses to assess the sensitivity of our results to assumptions. First,

we applied the method to the whole island data without geographical stratification, instead of differentiating Central and Outer communes in the main analysis. Second, we ran the stratified analysis by considering that, in the serological data, samples were IgG positive when the OD ratio was >2.5, rather than >3 in the main analysis.

**Reporting summary**. Further information on research design is available in the Nature Portfolio Reporting Summary linked to this article.

## Results

**Cases reported to the surveillance system**. As previously described[16,20], a RVF outbreak was declared in Mayotte with a total of 143 human cases reported between week 2018–47 (November 2018) and week 2019–30 (July 2019), including 137 who were ≥15 years old. The epidemic peaked early February 2019 (on week 2019–06) with 20 reported human cases (Fig. 1 and Supplementary Data 1). Geographical information was missing for 6 reported cases aged ≥15 years old.

**Serological data**. Mayotte seroprevalence survey was led in humans from week 2018–49 (December 2018) to week 2019–21 (May 2019) (Fig. 1). The distribution of the values of OD ratio obtained from the 2853 collected sera is displayed in Supplementary Fig. S1, suggesting that the cut-off of 3 correctly discriminated positive and negative samples. Using this cut-off, 254 out of 2854 samples were RVFV IgG positive. The positivity of samples depended on the timing of their collection, with a lower positivity around the beginning of the outbreak (Fig. 3 and Supplementary Data 1), hence justifying the need for a model reconstructing the temporal evolution of RVF seroprevalence and attack rate in humans. Indeed, in Central communes (resp. Outer communes), the observed IgG seroprevalence was 8.4% (19/225) (resp. 2.9% (2/69)) in the first three weeks of sampling compared to 12.2% (25/205) (resp. 8.2% (19/233)) in the last three analyzed weeks of sampling (Supplementary Data 1). Among sampling weeks included in the analysis, the maximum observed seroprevalence was 17.5% (11/63) on week 2019–14 (April 2019) in Central communes and 19.5% (8/41) on week 2019–08 (February 2019) in Outer communes.

**Estimates of seroprevalence, attack rate and reporting fraction**. Our model succeeded in fitting both case count data and seroprevalence data for each subpopulation, as most of the observed data were in the model 95% prediction intervals (Fig. 3B, C, E, F).

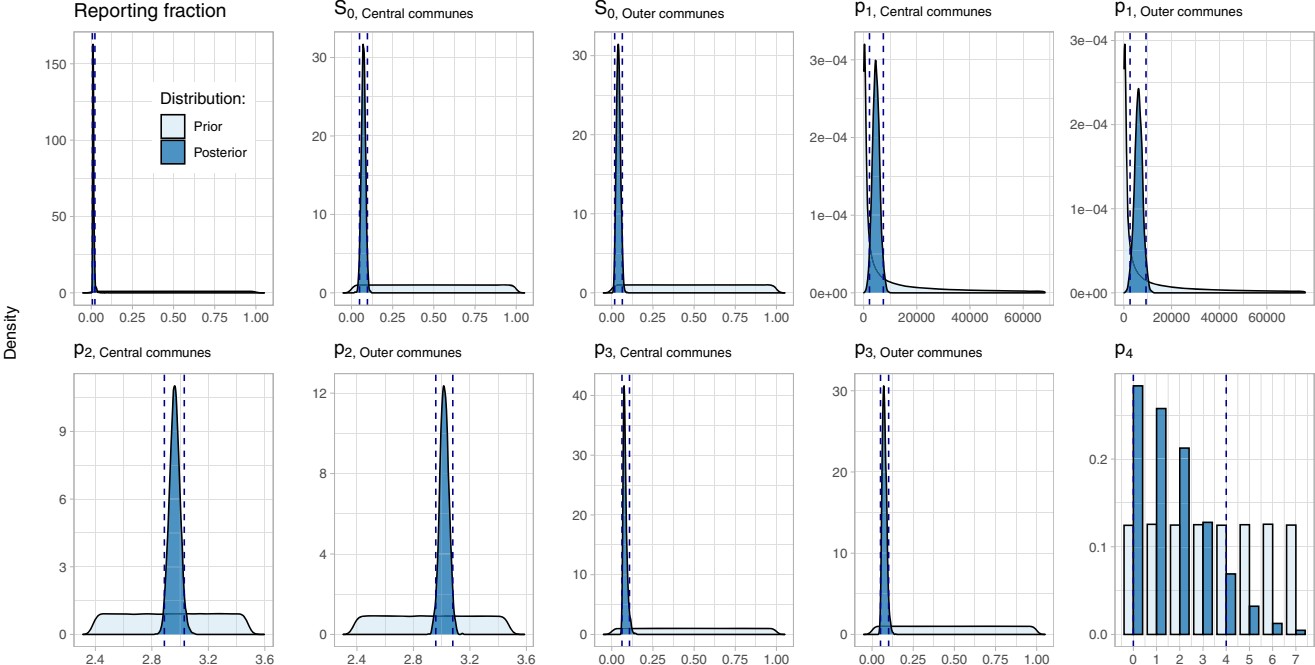

**Fig. 2 Prior (light blue) and posterior (dark blue) distributions of the model's parameters estimated by the Markov Chain Monte Carlo algorithm.** For each parameter, dashed vertical lines represent the posterior 95% credible interval (highest posterior density interval).

We estimated that a total of 10,797 people over 15 years old (95% Credible Interval, CrI: 4,728–16,127) were infected by RVFV in Mayotte during 2018–2019 epidemic. This represented 7.5% (3.3%–11.2%) of the total population of this age. The reporting fraction during the outbreak was estimated at 1.2% (0.67%–2.2%). Furthermore, Central and Outer communes were similarly affected overall, with 6.8% (3.2%–10.9%) and 8.1% (3.4%–12.2%) of their population over 15 years old infected, respectively (Fig. 2 and Supplementary Table S2).

The estimated IgG seroprevalence increased from 5.5% (3.6%–7.7%) before the outbreak to 12.9% (10.4%–16.3%) thereafter, for the whole island. Split by place of residence, we estimated an increase of the estimated IgG seroprevalence from 7.2% (4.9%–9.7%) to 13.9% (11.3%–17.3%) in Central communes, and from 4.0% (1.8%–6.4%) to 12.0% (8.5%–15.5%) in Outer communes.

We also estimated a delay of 1 week (0–4 weeks) between infection and the detectability of IgG antibodies in humans (Fig. 2 and Supplementary Table S2).

When simulating the model using posterior estimates of parameters (Fig. 3), the peak in human infections—both reported and unreported to the surveillance system—was predicted to occur on median on week 2019–06 in Central communes and on week 2019-07 in Outer communes (February 2019 in both areas).

**Sensitivity analyses**. In both sensitivity analyses, we obtained estimates similar to the baseline analysis, as detailed in Supplementary Notes 2, 3. In particular, the reporting fraction was estimated to 1.3% (0.70%–2.4%) and 1.2% (0.66%–2.2%) in the analyses using unstratified data and using a different serological cut-off, respectively.

## Discussion
In this analysis, we fitted a model to both serological and surveillance data collected during the 2018–2019 RVF outbreak in

Mayotte, which allowed us to estimate key parameters of the epidemic.

We estimated that 10,797 persons (aged over 15 years) were infected by RVFV throughout the 2018–2019 outbreak in Mayotte. This represented 7.5% of the population of this age on the island. However, only an estimated 1.2% of these infections were reported, despite the presence of a syndromic surveillance system on the island. This suggests that a large part of human cases were not diagnosed, although our study cannot determine whether the reason was because they presented no or mild symptoms, because they did not reach medical care while symptomatic, or both. Consistently with the first hypothesis, the proportion of RVFV infected humans with no or mild symptoms is generally considered to be >90%[3,8]. In the future, including a question about recent illness in seroprevalence surveys may help to disentangle the factors of under-reporting. Furthermore, strengthening surveillance at the interface between animal and human health sectors might allow to increase the reporting fraction and to detect potential incursions of RVFV in the island as early as possible, in order to implement control measures in a cost-effective way if needed[1,16].

The estimated IgG seroprevalence in people of Mayotte was 5.5% (95% CrI 3.6%–7.7%) just before the outbreak, as compared to the 3.5% (2.6%–4.8%) found in 2011[15]. Although we cannot rule out that a small number of cryptic RVFV infections in humans may have occurred[36,37], this result suggests that the circulation of the virus was negligible on the island between these dates. Moreover, no human RVF case was confirmed on the island between 2011 and 2018, and RVFV IgG seroprevalence in ruminants of Mayotte decreased continuously between 2011 and early 2018[38]. The small difference between the 2011 and the present studies may be due to difference in the sampled populations: contrary to our study, 5–14 years old were included in the 2011 survey and had a seroprevalence of 0.4%[15].

After the outbreak, in 2019, RVFV IgG seroprevalence in humans was estimated at 12.9% (10.4%–16.3%). This result is consistent with a recent review which reported that, across published seroprevalence studies, 12.6% of samples collected in

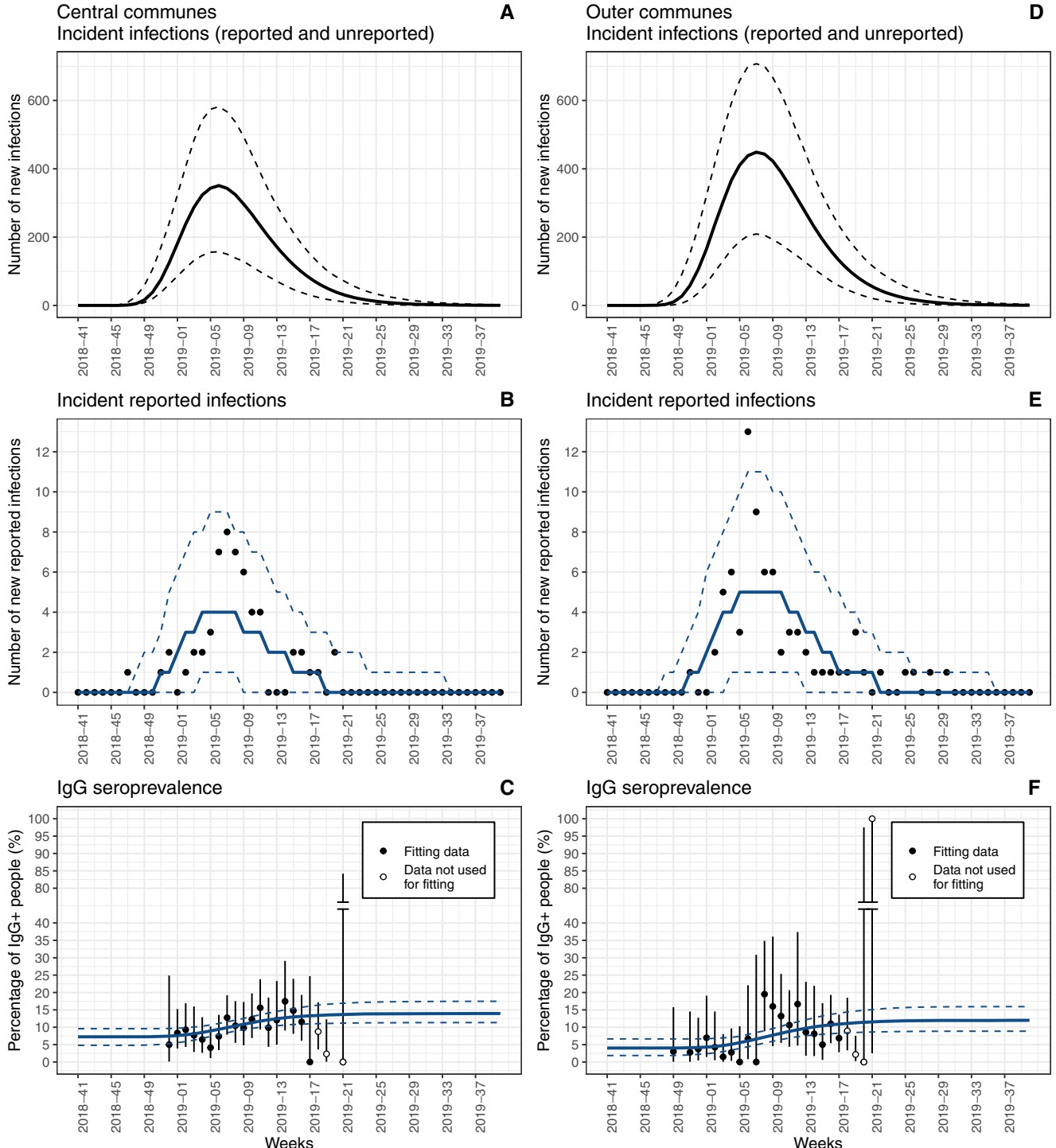

**Fig. 3 Time course of the RVF outbreak in humans.** Predicted number of weekly incident human infections, reported or unreported to the surveillance system (**A**, **D**), number of incident human infections reported to the surveillance system (**B**, **E**), and IgG seroprevalence in humans (**C**, **F**), in the population over 15 years old. **A–C** Represent Central Communes while **D–F** represent Outer Communes of Mayotte. Lines (solid and dashed) are model predictions (median and 95% prediction interval respectively, 5000 repetitions of the model). Dots are observed data (number of reported cases (**B**, **E**) and proportion of IgG seropositive tests (**C**, **F**)). Vertical bars (**C** and **F**) are 95% confidence interval for the proportion (Clopper-Pearson method). Serological data from week 2019–18 (early May 2019) onwards were not used for model fitting, because the representativeness of the sampled population was compromised (see Methods). In **C** and **F**, for visualization purposes, the *Y*-axis is cut between 40 and 80.

humans in the year following a RVF outbreak were positive for RVFV antibodies[3]. Even assuming IgG antibodies confer long-term protection against infection, this proportion would not prevent a hypothetical large outbreak of RVF in people of Mayotte in the future.

The estimated pre-epidemic seroprevalence was higher in Central (7.2%) than in Outer communes (4.0%), reflecting a higher exposure to RVFV in the past for people living in this area. This may be explained by an average higher proximity of these people to infected livestock animals in the past. Indeed, direct

contacts[15] and a closer spatial proximity[30] with food-producing animals have been identified as increasing the risk of infection. However, the proportion of people infected during the 2018–2019 outbreak was overall similar in both areas, probably as a result of the wide spread of RVFV among livestock populations of the whole island. Yet, the peak of human infections in Central communes was determined to occur 1 week ahead that in Outer communes, possibly reflecting the timing of the diffusion of RVFV in livestock, globally affecting Central before Outer communes[30].

In Outer communes, despite large confidence intervals, weekly seroprevalence data seemed to show a decrease starting on week 2019–08 (February 2019), which might be attributable to reducing levels of RVFV IgG antibodies in people that were infected earlier in the outbreak. Nevertheless, no comparable decreasing trend was observed in Central communes, and the sensitivity analysis performed with a lower serological cut-off did not result in different model estimates. In addition, IgG antibodies are generally considered to persist for several years[31,39], which makes their decline during the time of our study unlikely.

We estimated the period between infection by RVFV and the detectability of IgG antibodies to be 1 week, although with a wide 95% credible interval (0–4 weeks). We may have under-estimated this duration, since we considered that the week of infection was the week of symptoms onset (or, when missing, the week of RT-PCR confirmation) in reported cases. However, this under-estimation is probably <1 week, as much as RVFV incubation time[2,7,8] and viremia[31–34]. Furthermore, our estimation is in line with the range of values reported by Ref. [2,7,23–25], giving weight to our results.

This study has some limitations. First, some variables such as the age, gender, place of birth or occupational contacts with livestock were not accounted for in the analysis. The reason is some of these data were not collected as part of the cases reporting (for the place of birth) or serology survey (for the occupational contacts with livestock) datasets. Moreover, in a previous survey led in Mayotte in 2011[15], RVFV seroprevalence was not significantly associated with the age (after 15 years old) and gender.

Second, our modelling study did not include animal data. Indeed, rather than mechanistically simulating RVFV spill-over from animals to humans as in[16], our objective was to combine two independent sources of data to assess the extent of 2018–2019 RVF outbreak in humans. Consequently, our model did not explore the mechanisms that led to a decreased number of human cases. Yet, a previous publication[16] showed the epidemic fade out very likely resulted from the depletion of susceptible animals by natural infection, thus reducing the spill-over to humans. In the future, our results will be useful to parameterize such mechanistic models.

Third, we assumed that the reporting fraction was constant over time, although it might have varied throughout the course of the outbreak. However, the testing of all patients with dengue-like symptoms and negative to other infections (described above) has been implemented since 2008, and it is reasonable to suppose that the surveillance system had a steady capacity to detect RVF cases who sought medical care.

Fourth, we made the assumption that the IgG detection technique in serum had a sensitivity and a specificity of 1. If the sensitivity was <1 and the specificity was unchanged, the estimated outbreak's attack rate would be higher, and therefore the reporting fraction would be lower. On the other hand, the specificity of 1 is justified by the fact no other phlebovirus is known to circulate in this geographical area, preventing serological cross-reactivity with other viruses.

To conclude, combining incidence and seroprevalence data into a model, we estimated pre- and post-outbreak seroprevalence levels and reconstructed the true attack rate. This allowed us to provide the first estimate of RVF case reporting fraction during an epidemic, a key epidemiological parameter[40] which has rarely been assessed for other important infectious diseases[41–43].

## Data availability
The source data for Fig. 1 is in Supplementary Data 1.

## Code availability
Analyses were performed using R version 4.2.0. The code reproducing this article is available from https://doi.org/10.5281/zenodo.7343566[44].

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

## Acknowledgements

We are thankful to Marion Fleury, Jean-Baptiste Richard, Jean-Louis Solet, Laurent Filleul, Delphine Jezewski-Serra and Julie Chesneau for participating in Unono Wa Maore project conception, to the "URPS Infirmiers Ocean Indien" for conducting blood samplings, and to Marie-Claire Paty and Henriette de Valk for their proofreading. This work was funded by internal resources of Santé Publique France.

## Author contributions

J.B. performed the modelling analysis with inputs from L.D., J.P., N.H., R.M. and H.N. and wrote the first version of the manuscript. G.A.D. and G.G. conducted the serological analysis of blood samples. F.P., Y.H. and H.N. were involved in the collection of the surveillance data in humans, and M.R. in the design of Unono Wa Maore survey. H.N. supervised the project. All authors discussed the analysis and revised the manuscript.

## Competing interests

The authors declare the following competing interest: Raphaëlle Métras is an Editorial Board Member for Communications Medicine, but was not involved in the editorial review or peer review, nor in the decision to publish this article. The other authors declare no competing interests.
