## [Peer Review File · Communications Medicine]

Reviewers' comments:

Reviewer #1 (Remarks to the Author):

In this study the authors aim to analyse the dynamics of an RVFV outbreak in Mayotte in 2018-19 through mathematical modelling of surveillance and serological data. The study demonstrates good evidence that the majority of infections were not reported by the surveillance system during the epidemic. This work is an excellent demonstration of the value of serological data in outbreak analysis and provides valuable information regarding the surveillance system.

The article is well written, the findings are convincing and novel so I support the publication of this article. Below are some suggested revisions and questions for clarification that I believe will improve the article, most of which are minor but some major points that need addressing before publication.

Major

1. At line 102, why was OD ratio > 3 chosen as the cut-off for IgG positive? Did you test sensitivity to this cut-off? Is it possible to include data on the distribution of the OD ratio from all samples to justify this cut-off?
2. Under the ethics statement, there is no mention of ethics approval from the host country or the author's institution which I would expect to be in place for this work.
3. At line 184, is it possible to summarise the serological data before the section on model results? What was the pre-epidemic seroprevalence in each region? What was the maximum seroprevalence and when? This would be useful context when interpreting the model results.
4. There is no discussion of why the epidemic peaked and declined, even though estimated seroprevalence is low. What led to the effective reproduction number falling below 1 in this epidemic? If this cannot be explained with the model used then this should be included as a limitation.
5. There appears to be a discrepancy between Figure 1 and Figure S1 for week 2019-21 which is $>30\%$ but does not appear in Figure 1. This data point is such an outlier it needs discussion in the paper.

Minor

1. In the introduction, is there Anything on age of infection in the introduction? Mostly a childhood disease?
2. Confused about the acronym shift between RVF and RVFV
3. On line 57, the statement that "This calls for the use of mathematical and statistical modelling approaches..." needs further justification and explanation of why mathematical modelling can fill this knowledge gap, especially why serological data can help.
4. At line 95, include a brief description of the sampling method to justify the claim that they are "representative".
5. At line 97, include previous estimates of delay between infection and detectable IgG antibodies. I know this is discussed well later in the paper, but it is pertinent information for the reader.
6. At line 129 and SM2, why did you split the country by Central/Outer regions? Is it because of differences in income/housing/proximity to animals? I appreciate the details of the regions

are in SM2 but without prior knowledge of Mayotte it is unclear what the difference between these regions are.

7. At line 152, was the start date of the epidemic fixed in the model at week 41-2018? Did you analyse whether changing this start date changed your findings as has been shown in other island epidemics (<https://doi.org/10.1038/s41467-021-21788-y>).

8. At line 173, how was the likelihood constructed for the MCMC? Were the surveillance and serological fits added together? More details in SM3 would be welcome.

9. The inconsistent terms used for dates is slightly confusing, sometimes month-year is used and other times it is year-week number. Perhaps add the month information when using week number to help?

10. At line 209 and Figure 2, how did you select parameter estimates to simulate the model? Did you use the individual parameter estimates from table S3 or the parameter set with the best fitting likelihood? Correlation between parameter estimates could influence the findings from either approach so it should be clarified in the methods.

11. At line 234, is it possible to discuss whether the unreported cases were asymptomatic or not tested? Did the serological study ask about recent illness?

12. At line 266, specify whether the credible interval (0-4) is days or weeks.

13. At line 284-5, there is an excellent discussion of the limitations of assuming that IgG detection specificity was 1, but the implications of assuming a sensitivity of 1 is not mentioned, could this be added?

Reviewer #2 (Remarks to the Author):

I think this is a really interesting paper and a well done analysis. I just have a few comments as detailed below.

Introduction

I found the link between the first two lines a little confusing. Suggest rephrasing.

Line 57: "This calls for.." I find this a little strong, suggest rephrasing.

Line 60- 62: It would be helpful to state where these studies have taken place.

Line 65: It would be helpful to have more information on the type study done to get to this estimate.

Line 67: Please be more specific about the dates of the seroprevalence survey

Methods:

What was assumed about the time from infection to reporting as a case? If no delay, then I wonder if D represents something slightly different and is the time from day of symptoms (even if not shown) to IgG detection.

Table 1: The description of p2 and 3 is not very helpful. Also the description in the text is not helpful. This requires more detail for the reader to understand.

Results:

It would be helpful to have a figure of the distributions of the priors and the posterior parameter estimates.

Discussion:

I understand the smaller sample sizes/un-representativeness of the data in the clear circles, however there does seem to be a bit of a downward trend in the seroprev data even without this. I think this is worth commenting on in the discussion along with any information available on the longevity of the IgG or the cut off used to determine seropositivity in the study.

General comment: I couldn't see the author roles, but are the researchers who undertook the lab work and the surveys authors on this manuscript?

Reviewer #3 (Remarks to the Author):

This study combined surveillance and seroprevalence data together to provide epidemiological insights into RVF. I thought it was very interesting and well written, and provides valuable insights into the transmission of RVF.

My only major comment is that the authors have not provided their code alongside the manuscript. This is very important for the reproducibility of the paper, and without it can be difficult to assess the statistical analysis performed. The decision not to provide code may be due to limitations with sharing the full dataset, however, it should be possible to provide code with aggregated data, or to create some simulated data. If not, the code should be made available even without any data.

One small point, in equation line 153, it is not clear to me what $F_i(S)$ is. What is (s) ?

Otherwise, I think this was an excellent paper, with interesting results and a nice discussion.

Reconstructing Mayotte 2018-19 Rift Valley Fever outbreak in humans by combining serological and surveillance data

Response to reviewers

Reviewer #1 (Remarks to the Author):

In this study the authors aim to analyse the dynamics of an RVFV outbreak in Mayotte in 2018-19 through mathematical modelling of surveillance and serological data. The study demonstrates good evidence that the majority of infections were not reported by the surveillance system during the epidemic. This work is an excellent demonstration of the value of serological data in outbreak analysis and provides valuable information regarding the surveillance system.

The article is well written, the findings are convincing and novel so I support the publication of this article. Below are some suggested revisions and questions for clarification that I believe will improve the article, most of which are minor but some major points that need addressing before publication.

We thank the reviewer for their careful review of the manuscript and their useful remarks. Responses to the different comments are written in blue below. Line numbers correspond to the documents (main manuscript and Supplementary material) with visible changes.

Major

1. At line 102, why was OD ratio > 3 chosen as the cut-off for IgG positive? Did you test sensitivity to this cut-off? Is it possible to include data on the distribution of the OD ratio from all samples to justify this cut-off?

The cut-off of 3 was chosen accordingly to the cut-off used in previous studies investigating human cases infected with Rift Valley Fever virus and other arboviruses (Denis et al, 2019 ; Peyrefitte et al, 2005 ; Tong et al, 2019), whose references are now clearly specified on line 116.

We added a sensitivity analysis where we run the analysis using a value of 2.5 for this cut-off, on line 223 for the Methods and on line 292 for the Results, with additional results in Supplementary material SM7. We found that changing the cut-off value to 2.5 does not qualitatively change our results.

Moreover, we now added the distribution of the OD ratio from all samples in Supplementary material SM2, Figure S1 (in linear and log scales). As now written on line 238 of the main manuscript, the cut-off of 3 appears to correctly discriminate two distinct groups of samples, classified as positive and negative samples.

2. Under the ethics statement, there is no mention of ethics approval from the host country or the author's institution which I would expect to be in place for this work.

On line 127, the Ethics statement is now completed with more information : "Unono Wa Maore research protocol was validated by the Committee for the Protection of Persons (CPP, no. 2017-

A02782-51), the French ethical committee for biomedical research, and complied with MR001 reference methodology (agreement from the National Commission for Informatics and Freedoms of 25 September 2018, no. 918233). Information on the survey objectives and consent forms were read with the participants. A written informed consent was obtained from all participants who were ≥ 15 years old. All methods were carried out in accordance with relevant guidelines and the Declaration of Helsinki. All samples and data were anonymized at the time of collection. Therefore, sample testing and data analysis were conducted anonymously.”

3. At line 184, is it possible to summarise the serological data before the section on model results? What was the pre-epidemic seroprevalence in each region? What was the maximum seroprevalence and when? This would be useful context when interpreting the model results.

The “serological data” section of the Results is now completed with more details on the observed IgG seroprevalence on line 244. To describe values of observed seroprevalence calculated from a large enough quantity of samples, we compare the three first weeks of sampling to the last three weeks of sampling included in the analysis. We also describe the week and value of the maximum observed seroprevalence in both geographical areas.

“Indeed, in Central communes (resp. Outer communes), the observed IgG seroprevalence was 8.4% (19/225) (resp. 2.9% (2/69)) in the first three weeks of sampling compared to 12.2% (25/205) (resp. 8.2% (19/233)) in the last three analyzed weeks of sampling (SM1). Among sampling weeks included in the analysis, the maximum observed seroprevalence was 17.5% (11/63) on week 2019-14 in Central communes and 19.5% (8/41) on week 2019-08 in Outer communes.”

4. There is no discussion of why the epidemic peaked and declined, even though estimated seroprevalence is low. What led to the effective reproduction number falling below 1 in this epidemic? If this cannot be explained with the model used then this should be included as a limitation.

Our modelling framework does not allow to determine the causes of the decline of the epidemic, although a previous publication has mechanistically explored the dynamics of RVFV infections in animals and humans during this outbreak. We now discuss it on line 362:

“Consequently, our model did not explore the mechanisms that led to a decreased number of human cases. Yet, a previous publication (14) showed the epidemic fade out very likely resulted from the depletion of susceptible animals by natural infection, thus reducing the spill-over to humans.”

5. There appears to be a discrepancy between Figure 1 and Figure S1 for week 2019-21 which is $>30\%$ but does not appear in Figure 1. This data point is such an outlier it needs discussion in the paper.

In the originally submitted manuscript, the data point (not used in the fit) corresponding to the observed proportion of 1 sample positive / 1 sample collected (i.e. 100% [2.5% ; 100%]) in Outer Communes on week 2019-21 was reported in Table S1 but was not shown in Figure 2F, to avoid creating an Y-axis 0-100 precluding a good visualization of the data used for fitting (black dots) and of simulation results (Figure 2F was cut at Y-value 40, as described in the legend).

For consistency, we now show all data points and adjusted the Y-axis so our visualization is not altered: Figures 3-C, 3-F, S2-C, S3-C and S3-F now include a 0-100 Y-axis cut between 40 and 80. The precision was added in the figures’ legend. The point in Figure S2 for week 2019-21 corresponds to the observed seroprevalence of 1 sample positive / 3 samples collected (33.3% [0.8% ; 90.6%]) for this week in both geographical areas combined : 1/1 in Outer communes and 0/2 in Central communes (see Table S1).

Minor

1. In the introduction, is there Anything on age of infection in the introduction? Mostly a childhood disease?

On line 47, the precision “numerous spill-over human cases **(of all ages)**” was added, along with an additional reference (WHO, 1998) showing that the RVFV can infect people of all ages during outbreaks.

2. Confused about the acronym shift between RVF and RVFV

The acronym “RVF” (Rift Valley Fever) designates the disease, while “RVFV” (Rift Valley Fever Virus) designates the virus causing the disease. On line 44, we now wrote explicitly: “Rift Valley Fever virus (RVFV)”. On lines 57, 95, 165 and 322, we corrected “RVF” with “**RVFV**” to keep consistence throughout the text.

3. On line 57, the statement that “This calls for the use of mathematical and statistical modelling approaches...” needs further justification and explanation of why mathematical modelling can fill this knowledge gap, especially why serological data can help.

On line 60, we developed as follows: “In this context, mathematical and statistical models can be of interest to investigate the true burden of infection combining incomplete surveillance data with other sources of data (7,8). In particular, serological data have the advantage to provide biological markers of both symptomatic and asymptomatic previous infections”.

4. At line 95, include a brief description of the sampling method to justify the claim that they are “representative”.

On line 103, details are now added regarding Unono Wa Maore survey design:

“Briefly, dwellings were randomly drawn from the register of localized buildings, a database containing housing addresses (22). Within each participating household, up to three persons aged over 15 years old were then randomly selected. The characteristics of the surveyed population, notably in terms of age and sex, were comparable to the 2017 Mayotte general census (21).”

5. At line 97, include previous estimates of delay between infection and detectable IgG antibodies. I know this is discussed well later in the paper, but it is pertinent information for the reader.

On line 110, we added details on estimates of the delay to the detectability of RVFV IgG antibodies that were reported in the literature:

“These antibodies have been reported to be increasingly detectable from 8-10 days after RVFV infection in previous publications (2,7,22), or from 6-17 days after symptoms onset in others (23,24).”

6. At line 129 and SM2, why did you split the country by Central/Outer regions? Is it because of differences in income/housing/proximity to animals? I appreciate the details of the regions are in SM2 but without prior knowledge of Mayotte it is unclear what the difference between these regions are.

The stratification of the island into Central and Outer communes results from a previous publication (Kim et al., 2018) that has analysed the characteristics of the livestock movement network across the island. Central communes exhibited a more intense movement pattern than outer communes, and therefore affected the spread of RVFV in the livestock populations of Mayotte, and the spill-over to humans (Kim et al., 2018, 2021). More details are now added on line 152 of the main manuscript, and on line 74 of the Supplementary material (SM3).

7. At line 152, was the start date of the epidemic fixed in the model at week 41-2018? Did you analyse whether changing this start date changed your findings as has been shown in other island epidemics (<https://doi.org/10.1038/s41467-021-21788-y>).

The epidemic starting date, i.e. the week t from which $F_i(t) \geq 1$, is not fixed. Instead, parameters $p_{2,i}$ and $p_{3,i}$ (that we estimated) determine the shape of the epidemic curve, including the starting date. This precision was added on line 174.

Week 41-2018 only represents a lower bound of this starting date, because the function F_i (the epidemic curve) is not defined earlier in time. When simulating the fitted model, we see that the model does not tend to estimate a starting date earlier than week 46-2018. Indeed, in model simulations (see predictions' median and 95% intervals in Figures 3A and 3D), $F_i(t) < 1$ until that week. ($F_i(t)$ is mathematically always >0 but is very close to 0 during the first weeks.)

8. At line 173, how was the likelihood constructed for the MCMC? Were the surveillance and serological fits added together? More details in SM3 would be welcome.

The log-likelihoods of the “number of reported cases” and “number of IgG positive samples” components of the model for all weeks and for both geographical areas were indeed summed together. This precision was added on line 120 of the Supplementary Material (SM4).

9. The inconsistent terms used for dates is slightly confusing, sometimes month-year is used and other times it is year-week number. Perhaps add the month information when using week number to help?

To keep consistence, the month information was added on lines 120, 211, 212, 237, 238 and 286.

10. At line 209 and Figure 2, how did you select parameter estimates to simulate the model? Did you use the individual parameter estimates from table S3 or the parameter set with the best fitting likelihood? Correlation between parameter estimates could influence the findings from either approach so it should be clarified in the methods.

Regarding model simulations, we computed 5,000 repetitions of the model. For each repetition, we randomly selected an iteration of the posterior chains and used the set of parameters at this iteration. This precision on model simulations was added on line 212.

11. At line 234, is it possible to discuss whether the unreported cases were asymptomatic or not tested? Did the serological study ask about recent illness?

It was not possible to determine whether the unreported cases were asymptomatic or not tested, because the seroprevalence survey did not ask about recent illness. This point is now developed in line 303: “This suggests that a large part of human cases were not tested, although our study cannot determine whether the reason was because they presented no or mild symptoms, because they did not seek medical care while symptomatic, or both. Consistently with the first hypothesis, the proportion of RVFV infected humans with no or mild symptoms is generally considered to be $>90\%$ (3,8). In the future, including a question about recent illness in seroprevalence surveys may help disentangling the factors of under-reporting.”

12. At line 266, specify whether the credible interval (0-4) is days or weeks.

This was added on lines 266 and 346: “although with a wide 95% credible interval (0 - 4 **weeks**).”

13. At line 284-5, there is an excellent discussion of the limitations of assuming that IgG detection specificity was 1, but the implications of assuming a sensitivity of 1 is not mentioned, could this be added?

On line 372, the discussion was developed as follows: “If the sensitivity was lower than 1 and the specificity was unchanged, the estimated outbreak’s attack rate would be higher, and therefore the reporting fraction would be lower. On the other hand, the specificity of 1 is justified by the fact no other phlebovirus is known to circulate in this geographical area, preventing serological cross-reactivity with other viruses.”

Reviewer #2 (Remarks to the Author):

I think this is a really interesting paper and a well done analysis. I just have a few comments as detailed below.

We thank the reviewer for their careful review of the manuscript and their useful remarks. Responses to the different comments are written in blue below. Line numbers correspond to the documents (main manuscript and Supplementary material) with visible changes.

Introduction

I found the link between the first two lines a little confusing. Suggest rephrasing.

On line 42, we rephrased as follows: “Rift Valley Fever (RVF) is a viral mosquito-borne disease affecting both food-producing animals and humans, reported in most parts of Africa and the Arabian Peninsula. In some regions, the enzootic reservoir of Rift Valley Fever virus (RVFV) may consist of domestic or wild animals”

Line 57: “This calls for..” I find this a little strong, suggest rephrasing.

On line 60, we rephrased as follows: “In this context, mathematical and statistical models can be of interest to investigate the true burden of infection combining incomplete surveillance data with other sources of data”.

Line 60- 62: It would be helpful to state where these studies have taken place.

On line 65, the sentence is now completed as follows: “In previous studies **led in various areas of Africa, of the Indian Ocean and of Western and Southern Asia**, serological data have been used [...]”.

Line 65: It would be helpful to have more information on the type study done to get to this estimate.

The 2011 study was a seroprevalence survey. On line 71, this precision was added: “In 2011, following a RVF outbreak in livestock in 2008-2010 (16), **a serological survey estimated** the IgG RVFV seroprevalence in the human population of Mayotte (over 5 years old) to be 3.5%”.

Line 67: Please be more specific about the dates of the seroprevalence survey

On line 73, the period of the seroprevalence survey is now more accurately specified: “At the same period, **between December 2018 and May 2019**, the French public health agency conducted a seroprevalence study”.

Methods:

What was assumed about the time from infection to reporting as a case? If no delay, then I wonder if D represents something slightly different and is the time from day of symptoms (even if not shown) to IgG detection.

The time between the week of infection and the week of symptoms onset in reported cases (or the week of RT-PCR confirmation when it was missing) was neglected because both RVFV incubation time and viremia are less than a week. This precision is now added on line 183. Moreover, we now added in the Discussion (on line 346) the following point:

“We may have under-estimated this period, since we considered that the week of infection was the week of symptoms onset (or the week of RT-PCR confirmation) in reported cases. However, this under-estimation is probably less than 1 week, as much as RVFV incubation time (2,7,8) and viremia (31–34). Furthermore, our estimation is in line with the range of values reported by (2,7,23–25), giving weight to our results.”

Table 1: The description of p2 and 3 is not very helpful. Also the description in the text is not helpful. This requires more detail for the reader to understand.

On line 174 and in Table 1, more details are now given on $p_{2,i}$ and $p_{3,i}$:

“ $p_{2,i}$ and $p_{3,i}$ determine the shape of the epidemic curve including its duration, its starting date (i.e. the week t for which $F_i(t) \geq 1$) and the date of its peak (i.e. the mode of the distribution).”

Results:

It would be helpful to have a figure of the distributions of the priors and the posterior parameter estimates.

A figure showing prior and posterior distributions of parameters has now been added on line 216.

Discussion:

I understand the smaller sample sizes/un-representativeness of the data in the clear circles, however there does seem to be a bit of a downward trend in the seroprev data even without this. I think this is worth commenting on in the discussion along with any information available on the longevity of the IgG or the cut off used to determine seropositivity in the study.

We thank the reviewer for raising this point. We added a paragraph in the discussion (on line 338) to discuss this decreasing trend in the data:

“In Outer communes, despite large confidence intervals, weekly seroprevalence data seemed to show a decrease starting on week 2019-08 (February 2019), which might be attributable to reducing levels of RVFV IgG antibodies in people that were infected earlier in the outbreak. Nevertheless, no comparable decreasing trend was observed in Central communes, and the sensitivity analysis performed with a lower serological cut-off did not result in different estimates. In addition, IgG antibodies are generally considered to persist for several years (31,39), which makes their decline during the time of our study unlikely.”

General comment: I couldn't see the author roles, but are the researchers who undertook the lab work and the surveys authors on this manuscript?

The researchers who undertook the lab work at the National Reference Laboratory for Arboviruses are indeed authors on the manuscript (Guillaume André Durand and Gilda Grard). One member of the “Unono Wa Maore” group that conceived the survey is author (Marc Ruello), and the others are now cited in the Acknowledgements (on line 399), along with the “URPS Infirmiers Ocean Indien” that conducted blood samplings.

Reviewer #3 (Remarks to the Author):

This study combined surveillance and seroprevalence data together to provide epidemiological insights into RVF. I thought it was very interesting and well written, and provides valuable insights into the transmission of RVF.

We thank the reviewer for their careful review of the manuscript and their useful remarks. Responses to the different comments are written in blue below. Line numbers correspond to the documents (main manuscript and Supplementary material) with visible changes.

My only major comment is that the authors have not provided their code alongside the manuscript. This is very important for the reproducibility of the paper, and without it can be difficult to assess the statistical analysis performed. The decision not to provide code may be due to limitations with sharing the full dataset, however, it should be possible to provide code with aggregated data, or to create some simulated data. If not, the code should be made available even without any data.

We thank the reviewer for raising this important point. The code was shared on GitHub at this link: https://github.com/JonathanBas/RVF_Mayotte. A code availability statement is now included in the manuscript (on line 388). The data (reported cases and serological data) is aggregated by week and geographical area in the Supplementary material (SM1).

One small point, in equation line 153, it is not clear to me what $F_i(S)$ is. What is (s) ?

In the equation line 188, “ (s) ” corresponded to the “week” index within the sum. To make it clearer, we change “ (s) ” by “ (w) ”. In this equation, we sum all values of $F_i(w)$ (the number of incident human infections on week w) with w ranging from 1 to $t-D$.

Otherwise, I think this was an excellent paper, with interesting results and a nice discussion.

REVIEWERS' COMMENTS:

Reviewer #1 (Remarks to the Author):

In this revised manuscript describing an outbreak of Rift Valley fever virus in Mayotte in 2018-19 the authors have included excellent additional information that addresses all comments from the original submission. This study provides good evidence that the majority of infections were not detected during the outbreak and provides estimates of key epidemiological parameters, notably the reporting rate and the post outbreak seroprevalence.

Figure S1 in particular is an excellent addition, as is the sensitivity analysis of the choice of cut off for seropositivity. All minor comments from my original review have been well addressed with additional detail in the manuscript and supplement. I support publication of this work.

Reviewer #2 (Remarks to the Author):

thanks for the careful attention to my comments.

Reviewer #3 (Remarks to the Author):

I am satisfied that my comments have been addressed.